# A Variational Dirichlet Framework for Out-of-Distribution Detection

## Abstract

With the recently rapid development in deep learning, deep neural networks have been widely adopted in many real-life applications. However, deep neural networks are also known to have very little control over its uncertainty for test examples, which potentially causes very harmful and annoying consequences in practical scenarios. In this paper, we are particularly interested in designing a higher-order uncertainty metric for deep neural networks and investigate its performance on the out-of-distribution detection task proposed by Hendrycks & Gimpel (2016). Our method first assumes there exists an underlying higher-order distribution $\mathbb{P}(z)$, which generated label-wise distribution $\mathbb{P}(y)$ over classes on the K-dimension simplex, and then approximate such higher-order distribution via parameterized posterior function $p_\theta(z|x)$ under variational inference framework, finally we use the entropy of learned posterior distribution $p_\theta(z|x)$ as uncertainty measure to detect out-of-distribution examples. However, we identify the overwhelming over-concentration issue in such a framework, which greatly hinders the detection performance. Therefore, we further design a log-smoothing function to alleviate such issue to greatly increase the robustness of the proposed entropy-based uncertainty measure. Through comprehensive experiments on various datasets and architectures, our proposed variational Dirichlet framework with entropy-based uncertainty measure is consistently observed to yield significant improvements over many baseline systems.

## 1 Introduction

Recently, deep neural networks (LeCun et al., 2015) have surged and replaced the traditional machine learning algorithms to demonstrate its potentials in many real-life applications like speech recognition (Hannun et al., 2014), image classification (Deng et al., 2009; He et al., 2016), and machine translation (Wu et al., 2016; Vaswani et al., 2017), reading comprehension (Rajpurkar et al., 2016), etc. However, unlike the traditional machine learning algorithms like Gaussian Process, Logistic Regression, etc, deep neural networks are very limited in their capability to measure their uncertainty over the unseen test cases and tend to produce over-confident predictions. Such over-confidence issue (Amodei et al., 2016; Zhang et al., 2016) is known to be harmful or offensive in real-life applications. Even worse, such models are prone to adversarial attacks and raise concerns in AI safety (Goodfellow et al., 2014; Moosavi-Dezfooli et al., 2016). Therefore, it is very essential to design a robust and accurate uncertainty metric in deep neural networks in order to better deploy them into real-world applications. Recently, An out-of-distribution detection task has been proposed in Hendrycks & Gimpel (2016) as a benchmark to promote the uncertainty research in the deep learning community. In the baseline approach, a simple method using the highest softmax score is adopted as the indicator for the model's confidence to distinguish in- from out-of-distribution data. Later on, many follow-up algorithms (Liang et al., 2017; Lee et al., 2017; Shalev et al., 2018; DeVries & Taylor, 2018) have been proposed to achieve better performance on this benchmark. In ODIN (Liang et al., 2017), the authors follow the idea of temperature scaling and input perturbation (Pereyra et al., 2017; Hinton et al., 2015) to widen the distance between in- and out-of-distribution examples. Later on, adversarial training (Lee et al., 2017) is introduced to explicitly introduce boundary examples as negative training data to help increase the model's robustness. In DeVries & Taylor (2018), the authors proposed to directly output a real value between [0, 1] as the confidence measure. The most recent paper (Shalev et al., 2018) leverages the semantic dense

representation into the target labels to better separate the label space and uses the cosine similarity score as the confidence measure.

These methods though achieve significant results on out-of-distribution detection tasks, they conflate different levels of uncertainty as pointed in Malinin & Gales (2018). For example, when presented with two pictures, one is faked by mixing dog, cat and horse pictures, the other is a real but unseen dog, the model might output same belief as {cat:34%, dog:33%, horse:33%}. Under such scenario, the existing measures like maximum probability or label-level entropy (Liang et al., 2017; Shalev et al., 2018; Hendrycks & Gimpel, 2016) will misclassify both images as from out-of-distribution because they are unable to separate the two uncertainty sources: whether the uncertainty is due to the data noise (class overlap) or whether the data is far from the manifold of training data. More specifically, they fail to distinguish between the lower-order (aleatoric) uncertainty (Gal, 2016), and higher-order (episdemic) uncertainty (Gal, 2016), which leads to their inferior performances in detecting out-domain examples. In order to resolve the issues presented by lower-order uncertainty measures,

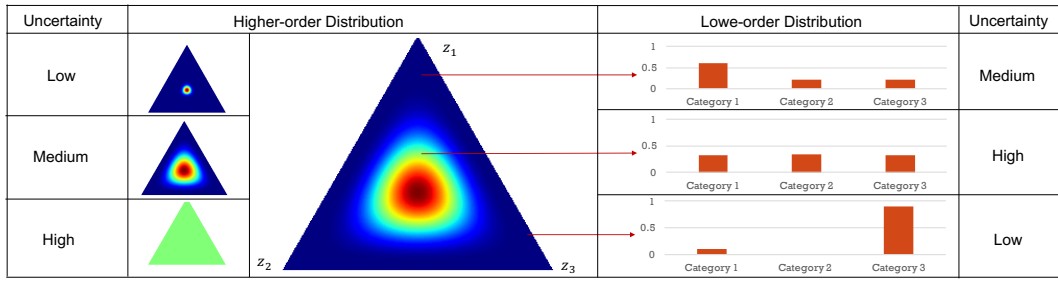

Figure 1: An intuitive explanation of higher-order distribution and lower-order distribution and their uncertainty measures.

we are motivated to design an effective higher-order uncertainty measure for out-of-distribution detection. Inspired by Subjective Logic (Jøsang, 2016; Yager & Liu, 2008; Sensoy et al., 2018), we first view the label-wise distribution $\mathbb{P}(y)$ as a K-dimensional variable $z$ generated from a higher-order distribution $\mathbb{P}(z)$ over the simplex $\mathbb{S}_k$, and then study the higher-order uncertainty by investigating the statistical properties of such underlying higher-order distribution. Under a Bayesian framework with data pair $D = (x, y)$, we propose to use variational inference to approximate such "true" latent distribution $\mathbb{P}(z) = p(z|y)$ by a parameterized Dirichlet posterior $p_\theta(z|x)$, which is approximated by a deep neural network. Finally, we compute the entropy of the approximated posterior for out-of-distribution detection. However, we have observed an overwhelming over-concentration problem in our experiments, which is caused by over-confidence problem of the deep neural network to greatly hinder the detection accuracy. Therefore, we further propose to smooth the Dirichlet distribution by a calibration algorithm. Combined with the input perturbation method (Liang et al., 2017; Krizhevsky & Hinton, 2009), our proposed variational Dirichlet framework can greatly widen the distance between in- and out-of-distribution data to achieve significant results on various datasets and architectures.

The contributions of this paper are described as follows:

- We propose a variational Dirichlet algorithm for deep neural network classification problem and define a higher-order uncertainty measure.
- We identify the over-concentration issue in our Dirichlet framework and propose a smoothing method to alleviate such problem.

## 2 MODEL

In this paper, we particularly consider the image classification problem with image input as $x$ and output label as $y$. By viewing the label-level distribution $\mathbb{P}(y) = [p(y = \omega_1), \cdots, p(y = \omega_k)]$ as a random variable $z = \{z \in \mathbb{R}^k : \sum_{i=1}^{k} z_i = 1\}$ lying on a K-dimensional simplex $\mathbb{S}_k$, we assume there exists an underlying higher-order distribution $\mathcal{P}(z)$ over such variable $z$. As depicted in Figure 1, each point from the simplex $\mathbb{S}_k$ is itself a categorical distribution $\mathbb{P}(y)$ over

different classes. The high-order distribution $\mathbb{P}(z)$ is described by the probability over such simplex $\mathbb{S}_k$ to depict the underlying generation function. By studying the statistical properties of such higher-order distribution $\mathbb{P}(z)$, we can quantitatively analyze its higher-order uncertainty by using entropy, mutual information, etc. Here we consider a Bayesian inference framework with a given dataset $D$ containing data pairs $(x, y)$ and show the plate notation in Figure 2, where $x$ denotes the observed input data (images), $y$ is the groundtruth label (known at training but unknown as testing), and $z$ is latent variable higher-order variable. We assume that the "true" posterior distribution is encapsulated in the partially observable groundtruth label $y$, thus it can be viewed as $\mathbb{P}(z) = p(z|y)$. During test time, due to the inaccessibility of $y$, we need to approximate such "true" distribution with the given input image $x$. Therefore, we propose to parameterize a posterior model $p_\theta(z|x)$ and optimize its parameters to approach such "true" posterior $p(z|y)$ given a pairwise input $(x, y)$ by minimizing their KL-divergence $D_{KL}(p_\theta(z|x)||p(z|y))$. With the parameterized posterior $p_\theta(z|x)$, we are able to infer the higher-order distribution over $z$ given an unseen image $x^*$ and quantitatively study its statistical properties to estimate the higher-order uncertainty.

In order to minimize the KL-divergence $D_{KL}(p_\theta(z|x)||p(z|y))$, we leverage the variational inference framework to decompose it into two components as follows (details in appendix):

$$D_{KL}(p_\theta(z|x)||p(z|y)) = -\mathcal{L}(\theta) + \log p(y) \tag{1}$$

where $\mathcal{L}(\theta)$ is better known as the variational evidence lower bound, and $\log p(y)$ is the marginal likelihood over the label $y$.

$$\mathcal{L}(\theta) = \mathbb{E}_{z \sim p_\theta(z|x)}[\log p(y|z)] - D_{KL}(p_\theta(z|x)||p(z)) \tag{2}$$

Since the marginal distribution $p(y)$ is constant w.r.t $\theta$, minimizing the KL-divergence $D_{KL}(p_\theta(z|x)||p(z|y))$ is equivalent to maximizing the evidence lower bound $\mathcal{L}(\theta)$. Here we propose to use Dirichlet family to realize the higher-order distribution $p_\theta(z|x) = Dir(z|\alpha)$ due to its tractable analytical properties. The probability density function of Dirichlet distribution over all possible values of the K-dimensional stochastic variable $z$ can be written as:

Figure 2: Plate Notation

$$Dir(z|\alpha) = \begin{cases} \frac{1}{B(\alpha)} \prod_{i=1}^{K} z_i^{\alpha_i - 1} & for \quad z \in \mathbb{S}_k \\ 0 & otherwise, \end{cases} \tag{3}$$

where $\alpha$ is the concentration parameter of the Dirichlet distribution and $B(\alpha) = \frac{\prod_i^K \Gamma(\alpha_i)}{\Gamma(\sum_i^k \alpha_i)}$ is the normalization factor. Since the LHS (expectation of log probability) has a closed-formed solution, we rewrite the empirical lower bound on given dataset $D$ as follows:

$$\mathcal{L}(\theta) = \sum_{(x,y) \in D} [\psi(\alpha_y) - \psi(\alpha_0) - D_{KL}(Dir(z|\alpha)||p(z))] \tag{4}$$

where $\alpha_0$ is the sum of concentration parameter $\alpha$ over K dimensions. However, it is in general difficult to select a perfect model prior to craft a model posterior which induces an the distribution with the desired properties. Here, we assume the prior distribution is as Dirichlet distribution $Dir(\hat{\alpha})$ with concentration parameters $\hat{\alpha}$ and specifically talk about three intuitive prior functions in Figure 3. The first uniform prior aggressively pushes all dimensions towards 1, while the *-preserving priors

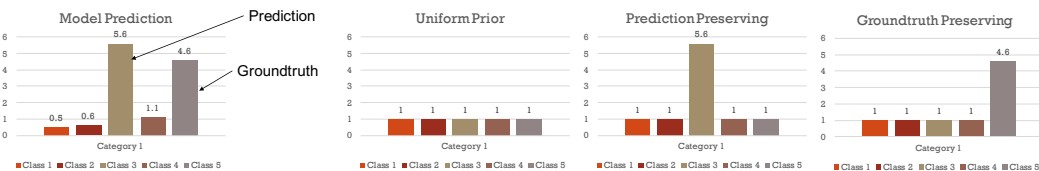

Figure 3: An intuitive explanation of different prior functions.

are less strict by allowing one dimension of freedom in the posterior concentration parameter $\alpha$. This is realized by copying the value from $k_{th}$ dimension of posterior concentration parameter $\alpha$ to the

uniform concentration to unbind $\alpha_k$ from KL-divergence computation. Given the prior concentration parameter $\hat{\alpha}$, we can obtain a closed-form solution for the evidence lower bound as follows:

$$\mathcal{L}(\theta) = \sum_{(x,y) \in D} [\psi(\alpha_y) - \psi(\alpha_0) - \log \frac{B(\hat{\alpha})}{B(\alpha)} - \sum_{i=1}^{k} (\alpha_i - \hat{\alpha}_i)(\psi(\alpha_i) - \psi(\alpha_0)) \quad (5)$$

$\Gamma$ denotes the gamma function, $\psi$ denotes the digamma function. We write the derivative of $\mathcal{L}(\theta)$ w.r.t to parameters $\theta$ based on the chain-rule: $\frac{\partial \mathcal{L}}{\partial \theta} = \frac{\partial \mathcal{L}}{\partial \alpha} \odot \alpha \cdot \frac{\partial f_\theta(x)}{\partial \theta}$, where $\odot$ is the Hardamard product and $\frac{\partial f_\theta(x)}{\partial \theta}$ is the Jacobian matrix. In practice, we parameterize $Dir(z|\alpha)$ via a neural network with $\alpha = f_\theta(x)$ and re-weigh the two terms in $\mathcal{L}(\theta)$ with a balancing factor $\eta$. Finally, we propose to use mini-batch gradient descent to optimize the network parameters $\theta$ as follows:

$$\frac{\partial \mathcal{L}}{\partial \alpha} = \sum_{(x,y) \in B(x,y)} [\frac{\partial [\psi(\alpha_y) - \psi(\alpha_0)]}{\partial \alpha} + \eta \frac{\partial D_{KL}(Dir(z|\alpha)||Dir(z|\hat{\alpha}))}{\partial \alpha}] \quad (6)$$

where $B(x, y)$ denotes the mini-batch in dataset $D$. During inference time, we use the marginal probability of assigning given input $x$ to certain class label $i$ as the classification evidence:

$$p(y = i|x) = \int_z p(y = i|z)p_\theta(z|x)dz = \frac{\alpha_i}{\sum_{j=1}^{k} \alpha_j} \quad (7)$$

Therefore, we can use the maximum $\alpha$'s index as the model prediction class during inference $\hat{y} = \arg\max_i p(y = i|x) = \arg\max_i \alpha_i$.

## 3 UNCERTAINTY MEASURE

After optimization, we obtain a parametric Dirichlet function $p_\theta(z|\alpha)$ and compute its entropy $E$ as the higher-order uncertainty measure. Formally, we write the such metric as follows:

$$E(\alpha) = -C(\alpha) = -\int_z z Dir(z|\alpha)dz = \log B(\alpha) + (\alpha_0 - K)\psi(\alpha_0) - \sum_{i}^{k} (\alpha_i - 1)\psi(\alpha_i) \quad (8)$$

where $\alpha$ is computed via the deep neural network $f_\theta(x)$. Here we use negative of entropy as the confidence score $C(\alpha)$. By investigating the magnitude distribution of concentration parameter $\alpha$ for in-distribution test cases, we can see that $\alpha$ is either adopting the prior $\alpha = 1.0$ or adopting a very large value $\alpha \gg 1.0$. In order words, the Dirichlet distribution is heavily concentrated at a corner of the simplex regardless of whether the inputs are from out-domain region, which makes the model very sensitive to out-of-distribution examples leading to compromised detection accuracy. In order to resolve such issue, we propose to generally decrease model's confidence by smoothing the concentration parameters $\alpha$, the smoothing function can lead to opposite behaviors in the uncertainty estimation of in- and out-of-distribution data to enlarge their margin.

**Concentration smoothing**     In order to construct such a smoothing function, we experimented with several candidates and found that the log-smoothing function $\hat{\alpha} = \log(\alpha + 1)$ can achieve generally promising results. By plotting the histogram of concentration magnitude before and after log-scaling in Figure 4, we can observe a very strong effect in decreasing model's overconfidence, which in turn leads to clearer separation between in- and out-of-distribution examples (depicted in Figure 4). In the experimental section, we detail the comparison of different smoothing functions to discuss its impact on the detection accuracy.

**Input Perturbation**     Inspired by fast gradient sign method (Goodfellow et al., 2014), we propose to add perturbation in the data before feeding into neural networks:

$$\hat{x} = x - \epsilon * sign(\nabla_x [\psi(\alpha_0) - \psi(\alpha_y)]) \quad (9)$$

where the parameter $\epsilon$ denotes the magnitude of the perturbation, and $(x, y)$ denotes the input-label data pair. Here, similar to Liang et al. (2017) our goal is also to improve the entropy score of any given input by adding belief to its own prediction. Here we make a more practical assumption that we have no access to any form of out-of-distribution data. Therefore, we stick to a rule-of-thumb value $\epsilon = 0.01$ throughout our experiments.

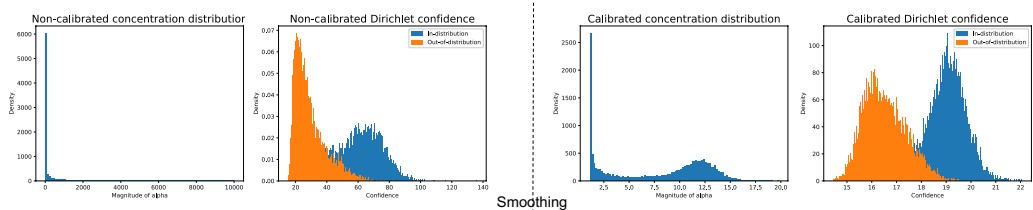

Figure 4: Concentration and confidence distribution before and after smoothing for CIFAR10 under VGG13 architecture with iSUN as out-of-distribution dataset.

| Dataset | Cross-Entropy | | | Ours | | |
|---|---|---|---|---|---|---|
| | VGG13 | WideResNet | ResNet-18 | VGG | WideResNet | ResNet-18 |
| CIFAR10 | 93.2 | - | 94.4 | 93.8 (+0.6) | - | 94.8 (+0.4) |
| CIFAR100 | - | 81.4 | - | - | 81.8 (+0.4) | - |
| SVHN | 96.8 | 96.7 | - | 95.9 (-0.9) | 96.1 (-0.6) | - |

Table 1: Classification accuracy of Dirichlet framework on various datasets and architectures.

**Detection**    For each input $x$, we first use input perturbation to obtain $\hat{x}$, then we feed it into neural network $f_\theta(\hat{x})$ to compute the concentration $\alpha$, finally we use log-scaling to calibrate $\alpha$ and compute $C(\hat{\alpha})$. Specifically, we compare the confidence $C(\hat{\alpha})$ to the threshold $\delta$ and say that the data $x$ follows in-distribution if the confidence score $C(\hat{\alpha})$ is above the threshold and that the data $x$ follows out-of-distribution, otherwise.

## 4    EXPERIMENTS

In order to evaluate our variational Dirichlet method on out-of-distribution detection, we follow the previous paper (Hendrycks & Gimpel, 2016; Liang et al., 2017) to replicate their experimental setup. Throughout our experiments, a neural network is trained on some in-distribution datasets to distinguish against the out-of-distribution examples represented by images from a variety of unrelated datasets. For each sample fed into the neural network, we will calculate the Dirichlet entropy based on the output concentration $\alpha$, which will be used to predict which distribution the samples come from. Finally, several different evaluation metrics are used to measure and compare how well different detection methods can separate the two distributions.

### 4.1    IN-DISTRIBUTION AND OUT-OF-DISTRIBUTION DATASET

These datasets are all available in Github [1].

- In-distribution: CIFAR10/100 (Krizhevsky & Hinton, 2009) and SVHN (Netzer et al., 2011), which are both comprised of RGB images of $32 \times 32$ pixels.
- Out-of-distribution: TinyImageNet (Deng et al., 2009), LSUN (Yu et al., 2015) and iSUN (Xiao et al., 2010), these images are resized to $32 \times 32$ pixels to match the in-distribution images.

Before reporting the out-of-distribution detection results, we first measure the classification accuracy of our proposed method on the two in-distribution datasets in  Table 1, from which we can observe that our proposed algorithm has minimum impact on the classification accuracy.

### 4.2    TRAINING DETAILS

In order to make fair comparisons with other out-of-distribution detectors, we follow the same setting of Liang et al. (2017); Zagoruyko & Komodakis (2016); DeVries & Taylor (2018); Shalev et al. (2018) to separately train WideResNet (Zagoruyko & Komodakis, 2016) (depth=16 and widening factor=8 for SVHN, depth=28 and widening factor=10 for CIFAR100), VGG13 (Simonyan & Zisserman, 2014), and ResNet18 (He et al., 2016) models on the in-distribution datasets. All models are

---

[1]https://github.com/ShiyuLiang/odin-pytorch

trained using stochastic gradient descent with Nesterov momentum of 0.9, and weight decay with 5e-4. We train all models for 200 epochs with 128 batch size. We initialize the learning with 0.1 and reduced by a factor of 5 at 60th, 120th and 180th epochs. we cut off the gradient norm by 1 to prevent from potential gradient exploding error. We save the model after the classification accuracy on validation set converges and use the saved model for out-of-distribution detection.

## 4.3 EXPERIMENTAL RESULTS

We measure the quality of out-of-distribution detection using the established metrics for this task (Hendrycks & Gimpel, 2016; Liang et al., 2017; Shalev et al., 2018). (1) FPR at 95% TPR (lower is better): Measures the false positive rate (FPR) when the true positive rate (TPR) is equal to 95%. (2) Detection Error (lower is better): Measures the minimum possible misclassification probability defined by $\min_\delta\{0.5P_{in}(f(x) \leq \delta) + 0.5P_{out}(f(x) > \delta)\}$. (3) AUROC (larger is better): Measures the Area Under the Receiver Operating Characteristic curve. The Receiver Operating Characteristic (ROC) curve plots the relationship between TPR and FPR. (4) AUPR (larger is better): Measures the Area Under the Precision-Recall (PR) curve, where AUPR-In refers to using in-distribution as positive class and AUPR-Out refers to using out-of-distribution as positive class.

| Model | OOD-Dataset | Method | FPR (TPR=0.95) | Detection Error | AUROC | AUPR In | AUPR Out |
|---|---|---|---|---|---|---|---|
| VGG13 CIFAR-10 | | Baseline | 43.8 | 11.4 | 94 | 95.5 | 91.5 |
| | | ODIN | 22.4 | 10.2 | 95.8 | 96.3 | 94.9 |
| | iSUN | BNN | 56.4 | 14.6 | 91.2 | 93.6 | 84.4 |
| | | Confidence | 16.3 | 8.5 | 97.5 | 98 | 96.9 |
| | | Ours | **10.9** | **6.9** | **98.0** | **98.4** | **97.7** |
| | | Baseline | 41.9 | 11.5 | 94 | 95.1 | 92.2 |
| | LSUN-resized | ODIN | 20.2 | 9.8 | 95.9 | 95.8 | 95.8 |
| | | BNN | 52.4 | 14.2 | 91.3 | 93.8 | 86.7 |
| | | Confidence | 16.4 | 8.3 | 97.5 | 97.8 | 97.2 |
| | | Ours | **9.9** | **6.6** | **98.1** | **98.4** | **97.9** |
| | | Baseline | 43.8 | 12 | 93.5 | 94.6 | 91.7 |
| | Tiny-ImageNet | ODIN | 24.3 | 11.3 | 95.7 | 95.9 | 95.9 |
| | | BNN | 53.7 | 16.9 | 90.2 | 91.9 | 82.6 |
| | | Confidence | 18.4 | 9.4 | 97 | 97.3 | 96.9 |
| | | Ours | **13.8** | **7.9** | **97.5** | **97.8** | **97.3** |
| VGG13 SVHN | | Baseline | 10 | 6 | 98 | 99.3 | 93.7 |
| | iSUN | ODIN | 1.6 | 2.95 | 99.5 | 99.8 | 98.8 |
| | | Confidence | 0.9 | 2.3 | 99.7 | 99.9 | 98.9 |
| | | Ours | **0.5** | **1.8** | **99.7** | **99.9** | **99.6** |
| | | Baseline | 9.4 | 5.7 | 98.1 | 99.3 | 94.3 |
| | LSUN-resized | ODIN | 1.4 | 2.6 | 99.6 | 99.7 | 99.1 |
| | | Confidence | 1 | 2.3 | 99.7 | 99.9 | 99 |
| | | Ours | **0.8** | **2.2** | **99.8** | **99.9** | **99.2** |
| | | Baseline | 11.4 | 6.2 | 97.8 | 99.2 | 93.7 |
| | Tiny-ImageNet | ODIN | 2.3 | 3.4 | 99.3 | 99.7 | 98.6 |
| | | Confidence | 1.5 | 2.8 | 99.5 | 99.8 | 98.7 |
| | | Ours | **1.4** | **2.3** | **99.7** | **99.8** | **99.2** |

Table 2: Experimental Results on VGG13 architecture, where Confidence refers to Learning Confidence algorithm (DeVries & Taylor, 2018), BNN refers to Bayesian Neural Network (Gal, 2016).

We report our VGG13's performance in Table 2 and ResNet/WideResNet's performance in Table 3 under groundtruth-preserving prior, where we list the performance of Baseline Hendrycks & Gimpel (2016), ODIN Liang et al. (2017), Bayesian Neural Network (Gal, 2016)[2], Semantic-Representation (Shalev et al., 2018) and Learning-Confidence (DeVries & Taylor, 2018). The results in both tables have shown remarkable improvements brought by our proposed variational Dirichlet framework. For CIFAR datasets, the achieved improvements are very remarkable, however, the FPR score on CIFAR100 is still unsatisfactory with nearly half of the out-of-distribution samples being

---

[2]We use the variational ratio as uncertainty measure to perform out-of-distribution detection, specifically, we forward Bayesian deep network 100 times for each input sample for Monte-Carlo estimation.

| Model | OOD-Dataset | Method | FPR (TPR=0.95) | Detection Error | AUROC | AUPR In | AUPR Out |
|---|---|---|---|---|---|---|---|
| ResNet18 CIFAR-10 | iSUN | Baseline | 52.6 | 13.6 | 92.4 | 94.6 | 88.9 |
| | | ODIN | 22.7 | 9.6 | 96.3 | 97.2 | 95 |
| | | Semantic | 21.5 | 9.2 | 96.3 | 97.1 | 94.3 |
| | | Ours | **13.8** | **8.4** | **97.1** | **97.4** | **96.8** |
| | LSUN-resized | Baseline | 50.2 | 12.3 | 93.1 | 94.8 | 90.8 |
| | | ODIN | 17.9 | 8.4 | 96.9 | 97.5 | 96.3 |
| | | Semantic | 23 | 14 | 96 | 96.7 | 94.8 |
| | | Ours | **12.1** | **6.3** | **97.4** | **97.4** | **97.5** |
| | Tiny-ImageNet | Baseline | 59 | 15.1 | 91.1 | 93.2 | 88.1 |
| | | ODIN | 32.1 | 11.2 | 94.9 | 95.8 | 93.6 |
| | | Semantic | 32.1 | 13.1 | 93.2 | 94.2 | 90.6 |
| | | Ours | **18.4** | **9.9** | **95.9** | **95.8** | **96.1** |
| WideResNet SVHN | iSUN | Baseline | 9.6 | 5.9 | 98 | 99.3 | 93.4 |
| | | ODIN | 1.1 | 2.7 | 99.6 | 99.8 | 99.1 |
| | | Ours | **0.8** | **1.8** | **99.8** | **99.9** | **99.5** |
| | LSUN-resized | Baseline | 9.5 | 5.8 | 98 | 99.3 | 94 |
| | | ODIN | 1.5 | 2.9 | 99.6 | 99.8 | 99 |
| | | Ours | **0.6** | **1.5** | **99.8** | **99.9** | **99.5** |
| | Tiny-ImageNet | Baseline | 10.6 | 6.1 | 97.8 | 99.2 | 93.6 |
| | | ODIN | 2.1 | 3.2 | 99.5 | 99.8 | 98.8 |
| | | Ours | **1.8** | **2.8** | **99.8** | **99.8** | **99.5** |

Table 3: Experimental results for ResNet architecture, where Semantic refers to multiple semantic representation algorithm (Shalev et al., 2018)

wrongly detected. For the simple SVHN dataset, the current algorithms already achieve close-to-perfect results, therefore, the improvements brought by our algorithm is comparatively minor.

## 4.4 ABLATION STUDY

In order to individually study the effectiveness of our proposed methods (entropy-based uncertainty measure, concentration smoothing, and input perturbation), we design a series of ablation experiments in Figure 5. From which, we could observe that concentration smoothing has a similar influence as input perturbation, the best performance is achieved when combining these two methods.

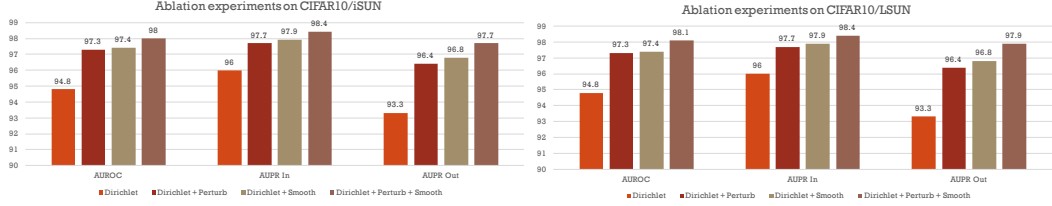

Figure 5: Ablation experiments for VGG13 architecture to investigate the impact of our proposed smoothing with CIFAR10 as in-distribution dataset and iSUN/LSUN as out-of-distribution dataset.

Here we mainly experiment with four different priors and depict our observations in Figure 6. From which, we can observe that the non-informative uniform prior is too strong assumption in terms of regularization, thus leads to inferior detection performances. In comparison, giving model one dimension of freedom is a looser assumption, which leads to generally better detection accuracy. Among these two priors, we found that preserving the groundtruth information can generally achieve slightly better performance, which is used through our experiments.

We also investigate the impact of different smoothing functions on the out-of-distribution detection accuracy. For smoothing functions, we mainly consider the following function forms: $\sqrt{x}$, $\sqrt[3]{x}$, $log(1 + x)$, $x$, $x^2$, $Sigmoid(x)$ and $SoftSign(x)$. Here we use $Sigmoid(x)$, $SoftSign(x)$ (range=$[0, 1]$) as baselines to investigate the impact of the range of smoothing function on the de-

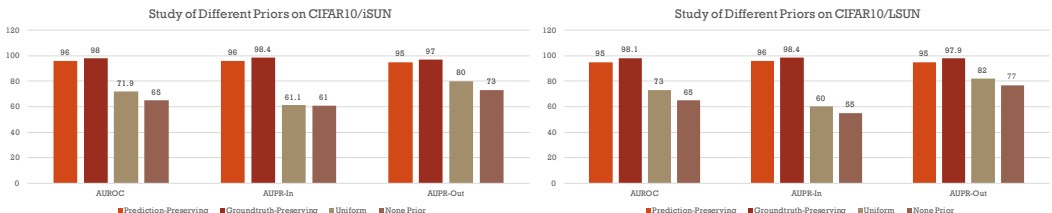

Figure 6: Impact of different prior distributions. The network architecture is VGG13 with CIFAR10 as in-distribution dataset and iSUN/LSUN as out-of-distribution dataset.

tection accuracy, and use $x, x^2$ as baselines to investigate the impact of compression capability of smoothing function on the detection accuracy. From Figure 7, we can observe that the first three smoothing functions greatly outperforms the baseline functions. Therefore, we can conclude that the smoothing function should adopt two important characteristics: 1) the smoothing function should not be bounded, i.e. the range should be $[0, \infty]$. 2) the large values should be compressed.

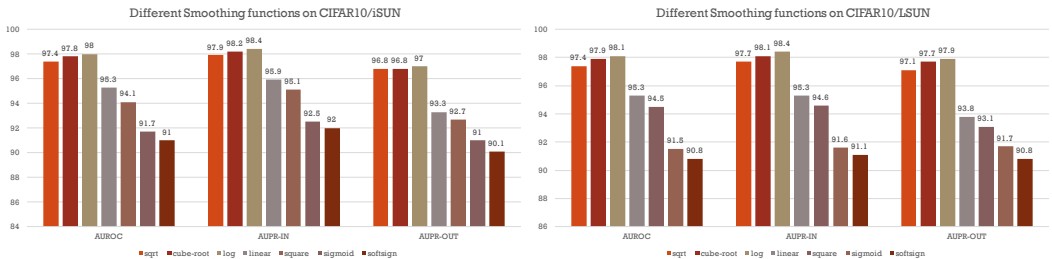

Figure 7: Impact of different smoothing functions. The network architecture is VGG13 with in-distribution CIFAR10 dataset and out-of-distribution iSUN/LSUN dataset.

## 5 RELATED WORK

The novelty detection problem (Pimentel et al., 2014) has already a long-standing research topic in traditional machine learning community, the previous works (Vincent & Bengio, 2003; Ghoting et al., 2008; Schlegl et al., 2017) have been mainly focused on low-dimensional and specific tasks. Their methods are known to be unreliable in high-dimensional space. Recently, more research works about detecting an anomaly in deep learning like Akcay et al. (2018) and Lee et al. (2017), which propose to leverage adversarial training for detecting abnormal instances. In order to make the deep model more robust to abnormal instances, different approaches like Bekker & Goldberger (2016); Xiao et al. (2015); Li et al. (2017); Lathuilière et al. (2018) have been proposed to increase deep model's robustness against outliers during training. Another line of research is Bayesian Networks (Gal & Ghahramani, 2016; 2015; Gal, 2016; Kingma et al., 2015), which are powerful in providing stochasticity in deep neural networks by assuming the weights are stochastic. However, Bayesian Neural Networks' uncertainty measure like variational ratio and mutual information rely on Monte-Carlo estimation, where the networks have to perform forward passes many times, which greatly reduces the detection speed.

## 6 CONCLUSION

In this paper, we aim at finding an effective way for deep neural networks to express their uncertainty over their output distribution. Our variational Dirichlet framework is empirically demonstrated to yield better results, but its detection accuracy on a more challenging setup like CIFAR100 is still very compromised. We conjecture that better prior Dirichlet distribution or smoothing function could help further improve the performance. In the future work, we plan to apply our method to broader applications like natural language processing tasks or speech recognition tasks.

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

# Appendices

## A  DERIVATION

Here we want to approximate the parameterized function towards to true distribution over latent variable $z$, which we write as:

$$
\begin{aligned}
& D_{KL}(p_\theta(z|x)||\mathbb{P}(z)) \\
=& D_{KL}(p_\theta(z|x)||p(z|y) \\
=& \int_z p_\theta(z|x) \log \frac{p_\theta(z|x)}{p(z|y)} dz \\
=& \int_z p_\theta(z|x) \log \frac{p_\theta(z|x)p(y)}{p(z,y)} dz \\
=& \int_z p_\theta(z|x) \log \frac{p_\theta(z|x)}{p(z,y)} dz + \log p(y) \\
=& \int_z p_\theta(z|x) \log \frac{p_\theta(z|x)}{p(z)p(y|z)} dz + \log p(y) \\
=& \int_z p_\theta(z|x) \log \frac{p_\theta(z|x)}{p(z)} dz - \int_z p_\theta(z|x) \log p(y|z) dz + \log p(y) \\
=& KL(p_\theta(z|x)||p(z)) - \mathop{\mathbb{E}}_{z \sim p_\theta(z|x)}[\log p(y|z)] + \log p(y) \\
=& - \big[ \mathop{\mathbb{E}}_{z \sim p_\theta(z|x)}[\log p(y|z)] - KL(p_\theta(z|x)||p(z))\big] + \log p(y) \\
=& - \mathcal{L}(\theta) + \log p(y)
\end{aligned}
\tag{10}
$$

## B  DETAILED DATASET

- CIFAR10/100 (in-distribution): The CIFAR-10 and CIFAR100 dataset (Krizhevsky & Hinton, 2009) consists of RGB images of $32 \times 32$ pixels. Each image is classified into 10/100 classes, such as dog, cat, automobile, or ship. The training split for both datasets is comprised of 50,000 images, while the test split is comprised of 10,000 images.

- SVHN (in-distribution): The Street View Housing Numbers (SVHN) dataset (Netzer et al., 2011) consists of colored housing number pictures ranging from 0 to 9. Images are also with a resolution of $32 \times 32$. The official training split is comprised of 73,257 images, and the test split is comprised of 26,032 images.

- TinyImageNet (out-of-distribution): The TinyImageNet dataset2 is a subset of the ImageNet dataset (Deng et al., 2009). The test set for TinyImageNet contains 10,000 images from 200 different classes for creating the out-of-distribution dataset, it contains the original images, downsampled to $32 \times 32$ pixels.

- LSUN (out-of-distribution): The Large-scale Scene UNderstanding dataset (LSUN) (Yu et al., 2015) has a test set consisting of 10,000 images from 10 different scene classes, such as bedroom, church, kitchen, and tower. We downsample LSUN's original image and create $32 \times 32$ images as an out-of-distribution dataset.
- iSUN (out-of-distribution): The iSUN dataset (Xiao et al., 2010) is a subset of the SUN dataset, containing 8,925 images. All images are downsampled to $32 \times 32$ pixels.

## C    EFFECTS ON KL-DIVERGENCE

Here we investigate the impact of KL-divergence in terms of both classification accuracy and detection errors. By gradually increasing the weight of KL loss (increasing the balancing factor $\eta$ from 0 to 10), we plot their training loss curve in Figure 8. With a too strong KL regularization, the model's classification accuracy will decrease significantly. As long as $\eta$ is within a rational range, the classification accuracy will become stable. For detection error, we can see from Figure 8 that adopting either too large value or too small value can lead to compromised performance. For the very small value $\eta \rightarrow 0$, the variational Dirichlet framework degrades into a marginal log-likelihood, where the concentration parameters are becoming very erratic and untrustworthy uncertainty measure without any regularization. For larger $\eta > 1$, the too strong regularization will force both in- and out-of-distribution samples too close to prior distribution, thus erasing the difference between in- and out-of-distribution becomes and leading to worse detection performance. We find that adopting a hyper-parameter of $\eta = 0.01$ can balance the stability and detection accuracy.

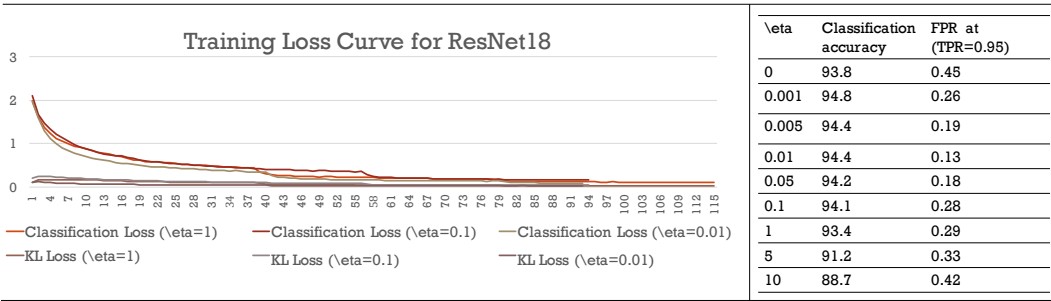

Figure 8: The training loss curve under ResNet18 on CIFAR10 dataset for different $\eta$ is demonstrated on the left side, the accuracy and out-of-distirbution detection results on the right side.

## D    RESULTS ON CIFAR100

Here we particularly investigate the out-of-distribution detection results of our model on CIFAR100 under different scenarios. Our experimental results are listed in Table 4.

| Model | OOD-Dataset | Method | FPR (TPR=0.95) | Detection Error | AUROC | AUPR In | AUPR Out |
|---|---|---|---|---|---|---|---|
| WideResNet CIFAR100 | iSUN | Baseline | 82.7 | 43.9 | 72.8 | 74.2 | 69.2 |
| | | ODIN | 57.3 | 31.1 | 85.6 | 85.9 | 84.8 |
| | | Semantic | 61.1 | 34.7 | 81.1 | 81.2 | 80.5 |
| | | Ours | **44.7** | **25.3** | **88.4** | **88.2** | **88.1** |
| | LSUN-resized | Baseline | 82.2 | 43.6 | 73.9 | 75.7 | 70.1 |
| | | ODIN | 56.5 | 30.8 | 86 | 86.2 | 84.9 |
| | | Semantic | 59.1 | 33.5 | 81.4 | 81.8 | 80.6 |
| | | Ours | **45.9** | **27.3** | **87.9** | **88.1** | **87.9** |
| | Tiny-ImageNet | Baseline | 79.2 | 42.1 | 72.2 | 70.4 | 70.8 |
| | | ODIN | 55.9 | 30.4 | 84.0 | 82.8 | 84.4 |
| | | Semantic | 59.3 | 31.5 | 82.8 | 81.3 | 81.3 |
| | | Ours | **51.2** | **28.6** | **86.1** | **84.9** | **86.1** |

Table 4: Experimental results for ResNet architecture, where Semantic refers to multiple semantic representation algorithm (Shalev et al., 2018)

