# OpenReview forum: "A Variational Dirichlet Framework for Out-of-Distribution Detection"
_ICLR.cc/2019/Conference_

### Official Review · AnonReviewer3 · 2018-10-28
**new method approximating the distribution of classification probability**

**Rating:** 6
**Confidence:** 3

**Review:**

This paper provides a new method that approximates the confidence distribution of classification probability, which is useful for novelty detection. The variational inference with Dirichlet family is a natural choice.

Though it is principally insightful to introduce the “higher-order” uncertainty, I do see the fundamental difference from the previous research on out-of-distribution detection (Liang, Lee, etc.). They are aimed at the same level of uncertainty.  Consider a binary classier, the only possible distribution of output y is Bernoulli- a mixture of Bernoulli is still Bernoulli.

In ODIN paper,  the detector contains both the measurement of the extent to which the largest unnormalized output of the neural network deviates from the remaining outputs (U1 in their notation) and another measurement of the extent to which the remaining smaller outputs deviate from each other (U2 in their notation).  In this paper, the entropy term has the same flavor as U2 part?

I am a little bit concerned with the VI approach, which introduces extra uncertainty.  I do not understand why there is another balancing factor eta in equation 6, which makes it no longer a valid elbo. Is the ultimate goal to estimate the exact posterior distribution of p(z) through VI, or purely some weak regularization that enforces uniformity?  Could you take advantage of some recent development on VI diagnostics and quantify how good the variational approximation is?

In general, the paper is clear and well-motivated, but I find the notation sometimes confusing and inconsistent. For example, the dependency on x and D is included somewhere but depressed somewhere else.  alpha_0 appears in equation 4 but it is defined in equation 7.

I am impressed by the experiment result that the new method almost always dominates best known methods, previously published in ICLR 2018. But I am not fully convinced why it works theoretically.  I would recommend a weak accept.

---

> ### Author Response · Authors · 2018-11-23
> **Thank you for your feedback**
>
> First of all, we really appreciate your useful suggestions. We have revised the paper to make it more clear and principled based on your suggestions.
>
> Q: "In ODIN ... U2 part"
> I agree with you, our smoothing algorithm has an inner connection with the temperature scaling algorithm. We think the complex entropy function can be decomposed into some function forms, let's say G function. By using log transform, the G function will have opposite behaviors for in- and out-of-distribution images. Thus, the in- and out-of-distribution examples are better separated.
>
> Q: "Confused by the VI approach"
> We have rewritten the model definition and show the derivation in Appendix. Here we briefly restate our model: what we are pursuing is the "true" unknown \mathbb{P}(z) = p(z|y), which receives ground truth label, we use p_{\theta}(z|x) to approximate such distribution through KL-divergence KL(p(z|x)||p(z|y)). We leverage variational inference to derive its lower bound (shown in appendix), and maximize this lower bound to minimize the KL-divergence. The prior is p(z), which does not consider any input, it's mainly used to induce desired properties in the approximated posterior distribution p_{\theta}(z|x). We experimented with three different kinds of priors (we updated in revision), enforcing uniformity actually leads to inferior performance. Therefore, we tried two other priors, namely "Groundtruth-preserving" and "Prediction-preserving", such prior can raise certain dimension in uniform [1,1, ..., 1] to be the same as the model prediction so as to preserve model's belief in a certain class. The preserving priors give the model one dimension of freedom and lead to generally better performance.
>
> Q: How good the approximation is?
> We are not sure about the which approximation you refer to, could you explain a bit more? The first error comes from the difference between p_{\theta}(z|x) and "true" unknown posterior p(z|y), and the second error comes from the difference between our computed ELBO and the "true" analytical ELBO. Our framework does not suffer from the second approximation error since the Dirichlet posterior can make ELBO adopt a closed-form solution.
>
> Q: Undefined alpha_0
> I have already fixed it, I define it in the first place it appears.
>
> Q: Theoretical ground, why it is better than previous ICLR papers?
> This is really a good question, I think there are mainly two reasons. First of all, we manage to separate the higher-level uncertainty from lower-level uncertainty, identifying the uncertainty sources can help to distinguish between "noisy in-domain example" (due to lower-level uncertainty) and "out-domain example" (due to higher-level uncertainty).  Secondly, our proposed smoothing algorithm can kind of alleviate the over-confidence issue in model training, thus making the model more robust to out-domain examples. These details are better demonstrated in the Introduction section in the current revision.

---

> > ### Comment · AnonReviewer3 · 2018-12-01
> > **Thanks for clarification**
> >
> > I notice you have made some changes to the paper. The new version is clearer.
> >
> > Prediction/truth-preserved Prior: Conceptually I am not sure if you still want to call that prior if it is constructed based on data.  But I think it makes sense that you want to maintain label-prediction while increasing/maximizing entropy.
> >
> > --"By investigating the magnitude distribution of concentration parameter α for in-distribution test cases, we can see that α is either adopting the prior α = 1.0 or adopting a very large value α ≫ 1.0. In order words, the Dirichlet distribution is heavily concentrated at a corner of the simplex regardless of whether the inputs"
> > If sometimes α can sometimes be adopted to be 1.0, why do you say it is *always* heavily concentrated at a corner?
> >
> > minor:
> > "Lowe-order" in Figure 1
> > "*-preserving priors" before figure 3
> > Figure 4: labels are too small to read

---

> > > ### Author Response · Authors · 2018-12-03
> > > **Thank you for your feedback**
> > >
> > > 1. My design is inspired by "Evidential Deep Learning to Quantify Classification Uncertainty" (equation 9). Maybe it is better to revise it back to the concentration "clipping" version so that the uniform prior is not dependent on the data.
> > >
> > > 2. Due to the regularization, most of the concentration parameters are around 1.0, while there are certain dimensions adopting extremely large values like 1000+, for examples, an image (label=3) could output its Dirichlet concentration parameter as [1.0, 1.1, 200, 1.0, 1500], the mode of such Dirichlet is [0, 0.00005, 0.11, 0, 0.88],  this distribution is extremely sharp at the corner (edge) between class 3 and class 5. Such over-confidence on the misclassified example makes the model very sensitive, by adopting a log smoothing, the distribution becomes [0.6, 0.7, 5.3, 0.6, 7.3], which is way less sharp than the previous one. Such confidence decreasing is demonstrated to have a stronger impact on the out-distribution examples than in-distribution examples, thus able to better separate these two input sources.
> > >
> > > 3. We will correct the minors in the revision.

---

### Official Review · AnonReviewer2 · 2018-10-31
**Not well motivated.**

**Rating:** 5
**Confidence:** 5

**Review:**

This paper proposes a new framework for out-of-distribution detection, based on variational inference and a prior Dirichlet distribution. The Dirichlet distribution is presented, and the way it is used withing the method is discussed (i.e. clipping, scaling, etc). Experiments on several datasets and comparison with the state of the art is reported and extensively discussed.

The motivation of the proposed approach is clear, and I agree with the attempt to regularize the network output. The choice of the Dirichlet distribution is quite natural, as each of its samples are the prior weights of a multinomial distribution. However, some other choices are less clear (clipping, scaling, decision, and the use of a non-informative prior). The overall inference procedure appears to be advantageous in the many experiments that the authors report (several datasets, and several baselines).

The first thought that came to my mind, is that out-of-distribution detection is to classification what outlier detection is to regression. Therefore, relevant and recent work on the topic deserves to be cited, for instance:
S. Lathuiliere, P. Mesejo, X. Alameda-Pineda and R. Horaud, DeepGUM: Learning Deep Robust Regression with a Gaussian-Uniform Mixture Model, In ECCV, 2018.

One thing that I found quite strange at first sight is the choice of clipping the parameters of the Dirichlet distribution. It is said that this is done in order to choose an appropriate prior distribution. However, the choice is not very well motivated, because what "appropriate" means is not discussed. So why do you clip to 1? What would happen if some of the alpha's go over 1? Is it a numerical problem, a modeling problem, a convergence issue?

I would also like the authors to comment on the use of a non-informative Dirichlet distribution within the KL-divergence. The KL divergence measures the deviation between the approximate a posteriori distribution and the true one. If one selects the non-informative Dirichlet distribution, this is not only a brutal approximation of the true posterior, but most importantly a distribution that does not depend on x, and that therefore cannot be truly called posterior.

It is also strange to take a decision based on the maximum alpha. On the contrary, the smallest alpha should be taken, since it is the one that concentrates more probability mass to the associated corner in the simplex.

Regarding the scaling function, it is difficult to grasp its motivation and effects. It is annouced that the aim of the smoothing function is to smooth the concentration parameters alpha. But in what sense? Why do they need to be smoothed? Is this done to avoid numerical/convergence problems? Is this a key part of the model? The same stands, by the way, for the form of the input perturbation.

The experiments are plenty, and I appreciated the sanity check done after introducing the datasets. However, I did not manage to understand why some methods appear in some tables and not in other (for example "Semantic"). I also have the feeling that the authors could have chosen CIFAR-100 in Table 2, since most of the metrics reported are quite high (the problems are not much challenging).

Regarding the parameter eta, I would say that its discussion right after Table 3 is not long enough. Specially, given the high sensitivity of this parameter, as reported in the Table of Figure 4. What is the overall interpretation of this sensitivity?

From a quantitative perspective, the results are impressive, since the propose methods systematically outperforms the baselines (at least the ones reported). However, since these baselines are not the same in all experiments, CIFAR-100 is not used, and the discussion of the results could be richer, I conclude that the experimental section is not too strong.

In addition to all my comments, I would say that the authors chose to exceed the standard limit of 8 pages. Even if this is allowed, the extra pages should be justified. I am affraid that there are many choices not well motivated, and that the discussion of the results is not informative enough. I am therefore not inclined to accept the paper as it is.

---

> ### Author Response · Authors · 2018-11-23
> **Thank you for your feedback**
>
> First of all, I would like to thank you for your constructive feedback. We revise the paper a lot based on your helpful suggestions.
>
> Q: Citing "outlier detection" papers:
> Thank you for your remind, I already cited a few papers in this domain in the related work in our revision.
>
> Q: Concentration Clipping problem:
> Sorry for the misunderstanding, this part in the previous version is really misleading. Therefore, I have already rewritten the introduction and model definition. Let me talk about the basic idea here, in order to compute the KL-divergence between Dirichlet posterior and prior KL(Dir(z|x)||p(z)), we need to design a prior for inducing desired property during optimization. The previous version proposes the "clipping" trick, assuming that a non-informative prior is given as [1,1,1,1,1] and our estimated posterior is [0.9, 1.2, 5, 1.1, 2.5], when the groundtruth label is 2, we want to give the model freedom in such groundtruth dimension (2) so that the model's belief in this dimension is not affected by the KL-regularization. Such a goal is realized by clipping [0.9, 1.2, 5, 1.1, 2.5] to [0.9, 1.2, 1, 1.1, 2.5] so that the second dimension is unbound from the KL-divergence computation.
> In the new version, we rewrite this part in a more principled way. As pointed in the new version, since it is really hard to find a "perfect" prior, our paper specifically picks three most intuitive possibilities (namely uniform, groundtruth-preserving and prediction-preserving) to see their results. Instead of clipping the concentration, we propose to raise the uniform's certain dimension, for example, for [0.9, 1.2, 5, 1.1, 6] and [1,1,1,1,1] with label=2, we change prior to [1,1,5,1,1] to allow freedom in second dimension (called "groundtruth-preserving"), while "prediction-preserving" changes prior to [1,1,1,1,6]. We performed ablation study to understand the impact of different priors on the final detection accuracy.
>
> Q: "I would also like the authors to comment on the use of a non-informative Dirichlet..."
> Sorry for the misunderstanding, the previous version is quite misleading, so we have dramatically rewritten the Model part to clearly state what is posterior, prior and "true" distribution. Let me briefly discuss it here, the "true" unknown distribution is described as "p(z|y)", which is the goal we need to approximate by minimizing KL(p(z|x)||p(z|y)) using posterior p(z|x). This KL-divergence optimization be further broken down into two parts E_{z} log(p(y|z)) + KL(p(z|x)||p(z)), the non-informative prior is used to described p(z) rather than p(z|y). Details for derivation is also demonstrated in the appendix eq10.
>
> Q:  Take a decision on maximum alpha
> The decision is based on the marginal probability p(y=i|x), which has an analytical solution. We demonstrate the inference process in eq 7 in our new revision. Hope you can take a look.
>
> Q: The smoothing function
> The smoothing function is to lower model's overconfidence on unseen samples so that the model can better detect the abnormal instances. For example, an unseen image whose label should be 2, but the model outputs [0.5, 0.1, 10, 100], the log smoothing can scale it to [0.4, 0.1, 2.3, 4.6], which greatly lowers model's confidence on its own prediction (y=3) while maintaining the other dimensions. Such smoothing function can increase the detection accuracy a lot, the details are shown in ablation study in Fig 5. In our revision, we also experimented with many other smoothing functions to investigate what is the essential property a smoothing function needs to make the model more robust against outliers, the results are shown in Fig 7. The same goes with input perturbation, which is used to enlarge the distance between normal instances and abnormal instances so that the model can better detect the abnormal ones.

---

> ### Author Response · Authors · 2018-11-23
> **Second Part due to comment limit**
>
> Q: "why some methods appear in some tables and not in other"
> Here we mainly copy the results reported in the original paper to ensure fairness, the previous papers use different architectures, therefore some results are missing. Specifically for CIFAR100, the previous papers like Learning-Confidence (Devries et al.), Adversarial Training (Lee et al.) and Deep Prior Network (Malinin et al.) did not report their results on such really challenging dataset. The most recently reported results are from ODIN (Liang et al.) and Semantic (Shalev et al.), therefore only these two baselines are compared in the more challenging CIFAR100 setting. The inferior performance on CIFAR100 is hard because there are more than 20% of misclassified test images, which is really hard to be distinguished from out-of-distribution images.
>
> Q: Regarding \eta
> Thank you for pointing this out, we vary \eta from 0.001-> 10 to see its influence on the final results, which is actually a pretty large range considering that 10/0.001 = 10000. In our new fig8, we show that varying \eta from 0.001 -> 0.1 (scale 100 times) will increase FPR from 0.13 -> 0.26, the change is actually not too dramatic. The ELBO objective is comprised of LHS expectation and RHS KL-divergence term. During training, the LHS objective converges at a different rate as the RHS KL-divergence, which leads to sensitivity problem. We experimented with some annealing algorithm to gradually decrease the KL-divergence, which can help erase such a sensitive problem, but it also introduces other complexity like annealing hyper-parameters, etc.
>
> Q: Page Limit
> Thank you for your remind, we already restructured the paper to fit exactly 8 pages, some parts are moved to the appendix. We really hope you can carefully read our revision again.

---

### Official Review · AnonReviewer1 · 2018-11-05
**Bayesian reasoning about DNN outcome**

**Rating:** 6
**Confidence:** 4

**Review:**

Summary
=========
The paper describes a probabilistic approach to quantifying uncertainty in DNN classification tasks.
To this end, the author formulate a DNN with a probabilistic output layer that outputs a multinomial over the
possible classes and is equipped with a Dirichlet prior distribution.
They show that their approach outperforms other SOTA methods in the task of out-of-distribution detection.

Review
=========
Overall, I find the idea compelling to treat the network outputs as samples from a probability distribution and
consequently reason about network uncertainty by analyzing it.
As the authors tackle a discrete classification problem, it is natural to view training outcomes as samples from
a multinomial distribution that is then equipped with its conjugate prior, a Dirichlet.

However, the model definition needs clarification. In the classical NN setting, I find it misleading
to speak of output distributions (here called p(x)). As the authors point out, NNs are deterministic function approximators
and thus produce deterministic output, i.e. rather a function f(x) that is not necessarily a distribution (although can be interpreted as a probability).
One could then go on to define a latent multinomial distribution over classes p(z|phi) instead that is parameterized by a NN, i.e. phi = f_theta(x).
The prior on p(phi) would then be a Dirichlet and consequently the posterior is Dirichlet as well.
The prior distribution should not be dependent on data x (as is defined below Eq. 1).

The whole model description does not always follow the usual nomenclature, which made it at times hard for me to grasp the idea.
For instance, the space that is modeled by the Dirichlet is called a simplex. The generative procedure, i.e. how does data y constructed from data x and the probabilistic procedure, is missing.
The inference procedure of minimizing the KL between approximation and posterior is just briefly described and could be a hurdle to understand, how the approach works when someone is unfamiliar with variational inference.
This includes a proper definition of prior, likelihood and resulting posterior (e.g. with a full derivation in an appendix).

Although the authors stress the importance of the approach to clip the Dirichlet parameters, I am still a bit confused on what the implications of this step are.
As I understood it, they clip parameters to a value of one as soon as they are greater than one.
This would always degrade an informative distribution to a uniform distribution on the simplex, regardless whether the parameters favor a dense or sparse multinomial.
I find this an odd behavior and would suggest, the authors comment on what they mean with an "appropriate prior". Usually, the parameters of the prior are fixed (e.g. with values lower one if one expects a sparse multinomial).
The prior then gets updated through the data/likelihood (here, a parameterized NN) into the posterior.

Clipping would also lead to the KL term in Eq. 3 to be 0 often times, as the Dir(z|\alpha_c) often degrades to Dir(z|U).

The experiments are useful to demonstrate the application and usefulness of the approach.
Outcome in table 3 could maybe be better depicted using bar charts, results from table 4 can be reported as text only, which would free up space for a more thorough model definition.

---

> ### Author Response · Authors · 2018-11-22
> **Thank you so much for your constructive feedback**
>
> First of all, I would like to thank you for your constructive feedback. Most of our revision is based on your helpful suggestions.
> 1. We totally agree with you on the NN setting and revise the introduction and model part, where we view the neural network only as a general approximation function to generate concentration parameters.
> 2. We follow the usual nomenclature in variational inference to revise the model definition part. In fig3, we specifically define the x,y,z and talk about their connections, the current prior distribution does not contain $x$ as input.
> 3. We write about the statistical process in the model section and define the three probabilities, and we demonstrate the derivation process in appendix eq10.
> 4. About the clipping part: sorry for the misunderstanding, the original version actually means clipping the groundtruth dimension of the predicted concentration to 1 while maintaining the rest dimensions. For example, for y=2 and concentration [0.6, 5, 1.2, 8], clipping will change it to [0.6, 1, 1.2, 8] so that the goundtruth dimension does not contribute to the KL-divergence with U=[1, 1, 1, 1]. The motivation of clipping is to give the model one dimension of freedom. In the new revision, we rewrite this part in a more principled way. Instead of clipping the concentration, we raise the prior uniform concentration in a certain dimension, for example [1, 1, 1, 1] will become [1, 5, 1, 1], which have the same goal of allowing one dimension of freedom in the concentration parameter. More importantly, we design different prior functions and perform ablation study to investigate their influences on the final results.
> 5. We change some tables into bar charts to better visualize the results.
>
> Again, thank you for your feedback and hope you can read our new revision.

---

### Public Comment · ~Andrey_Malinin1 · 2018-10-10
**Similar work**

Hello!

Your work is similar to our work which is due to appear at NIPS 2018 -  https://arxiv.org/pdf/1802.10501.pdf

Both our works parameterise a distribution over distributions using a DNN in order to derive measures of uncertainty in predictions for detection of out-of-distribution samples.

As far as I understand, the main differences between your work and ours are the following:

1. You interpret the the model to have latent variables which capture the distribution over distributions, while we interpret the model to be directly parameterizing a distribution over distributions. Though in the end the model architectures are essentially the same (DNN parameterises Dirichlet).
2. You train the model using ELBO using only in-domain data while we train the model using a contrastive KL-Divergence loss using in-domain and out-of-distribution data.
3. You use additional heuristics, like clipping and smoothing the alphas, in order to get a well-behaved model.
4. You investigate only the Differential Entropy of the Dirichlet as a measure of 'higher level uncertainty' while we investigate a range of uncertainty measures, derivable from distributions over distributions, which capture uncertainty in predictions due to different sources of uncertainty (data/distributional/model uncertainty).

Please correct me if I have misunderstood anything. I find your paper to be very interesting. Specifically, I find it impressive that you are able to achieve very good empirical results without out-of-distribution training data! Smoothing the alphas also sounds like a good idea :) .

Best Regards,
Andrey Malinin

---

> ### Author Response · Authors · 2018-10-11
> **Thank you for your insightful comments**
>
> Hi Andrey,
>
> Thank you so much for your careful and insightful comments. Before I respond to your comment, I hope to clarify that I wasn't aware of the existence of your paper before our submission, it was only a few days after the deadline that I found your paper through the reference of "https://openreview.net/forum?id=H1gh_sC9tm". After carefully reading your paper, I found it is really interesting and elegant, especially you have conducted comprehensive experiments on different uncertainty measures. I'm sorry that we didn't cite your paper in the current version, but we will definitely put your paper in our citation list in the future revision and compare against your results to gain a deeper understanding.
>
> Now I would like to answer your comments:
> 1. We indeed try to interpret the problem from two different angles, but we end up having the same architecture. I think it's probably due to the fact that Dirichlet seems to be the only "appropriate" choice for prior or posterior distribution (it has so many well-studied properties like entropy, variance, mean, etc).
> 2. I think this is one of the main differences between our papers. My method is based on ELBO, which only depends on the in-domain dataset and your method takes the contrastive loss, which depends on both in-domain and out-domain dataset (though the out-domain dataset can be synthesized).
> 3. Yes, clipping and smoothing are the two weapons (tricks) to make our method work without providing any out-domain dataset. Because the model never gets the chance to see the adversarial (out-domain) examples under our setting, it is very inclined to put extremely high confidence in its own beliefs. In order to alleviate such an issue, we are inspired by the distribution calibration technique in the statistical theory (http://scikit-learn.org/stable/modules/calibration.html) to use transform function to re-adjust the model's belief in a more rational range. We experimented with several different smoothing techniques and end up having the simple log(x+1) as our calibration function.
> 4. I really like your theoretical explanation for different uncertainty measures. In our framework, we actually experimented with some intuitive uncertainty measure like low-level entropy (over label) and max value, but their results turn out to be worse than our baseline (ODIN), therefore we just leave them out from the paper due to the page limit. Actually, we did compare against two different uncertainty measures (variational ratio (BNN) in table2, and evidential-Dirichlet uncertainty measure in table4). Anyway, I will list the results of other uncertainty measures in our future revision.
>
> Again, I'm really grateful for your helpful discussion!
>
> Best regards.

---

### Public Comment · (anonymous) · 2018-10-15
**Questions on ablation studies**

Hi, thanks for the nice paper.

I have a questions on ablation studies in Table 4.

>>We also observe that concentration smoothing is playing a more important role than input perturbation.

I think it unfair to compare "concentration smoothing" with "input perturbation". Is it necessary to design the Dirichlet+Perturbation experiment for each data sets ?

---

> ### Author Response · Authors · 2018-10-17
> **Thank you for your kindly remind**
>
> Thank you so much for your careful and insightful comments. I totally agree with you, the current ablation set up is a bit problematic. Therefore, we add a few experiments to demonstrate the following results:
>
> Dirichlet + Perturbation:
> Model        OOD       FPR   Detection Error
> VGG13       iSUN       16.8     8.5
> CIFAR10    LSUN       15.2    8.6
>                    Tiny-IM    20.3   9.9
>
> Dirichlet + Smoothing:
> Model        OOD       FPR   Detection Error
> VGG13       iSUN        14.4   7.9
> CIFAR10    LSUN        13.8   7.7
>                    Tiny-IM    18.9   9.1
>
> From the above observation, it's probably safer to claim that Dirichlet smoothing strategy is marginally more important than Input perturbation technique. Thank you for your kind remind, we will refine this part in the future revision.

---

### Author Response · Authors · 2018-11-27
**Summary of Revision**

We have submitted a revised manuscript and made the following modifications to address the reviewers' major concerns:

-- Add detailed explanation about what's lower-level uncertainty and what's higher-level uncertainty, why the higher-level uncertainty is better.
-- Add detailed model definition to follow variational inference nomenclature.
-- Add the derivation of our proposed evidence lower bound to Appendix.
-- Rewrite the prior part, the previous is to clip concentration in a certain dimension, the current version is to raise the uniform Dirichlet in a certain dimension. Though achieving the same effect, we believe such a change makes the paper easier to be understood.
-- Add more ablation studies to verify different prior functions and different smoothing functions to see their impact on the detection accuracy.
-- Update references according to Reviewer2 and Malinin.
-- Shorten the paper to fit the 8-page standard.
-- Fix many typos.

While limited by time in the response period, we do still plan to address *all* the reviewer’s other additional comments in future revisions. We also welcome any further feedback to improve this paper!

---

### Meta-Review · Area_Chair1 · 2018-12-12
**Arguable choices of parameters and the performance degradation issue**

**Confidence:** 4
**Recommendation:** Reject

**Metareview:**

The paper proposes a new framework for out-of-distribution detection, based on variational inference and a prior Dirichlet distribution.

The reviewers and AC note the following potential weaknesses: (1) arguable and not well justified choices of parameters and (2) the performance degradation under many classes (e.g., CIFAR-100).

For (2), the authors mentioned that this is because "there are more than 20% of misclassified test images". But, AC rather views it as a limitation of the proposed approach. The out-of-detection detection problem is a one or two classification task, independent of how many classes exist in the neural classifier.

In overall, the proposed idea is interesting and makes sense but AC decided that the authors need more significant works to publish the work.